# Knowledge, practices and seroprevalence of *Taenia* species in smallholder farms in Gauteng, South Africa

Nothando Altrecia Shongwe[1], Charles Byaruhanga[1], Pierre Dorny[2,3], Veronique Dermauw[2], Daniel Nenene Qekwana[4]*

1 Department of Veterinary Tropical Diseases, Faculty of Veterinary Science, University of Pretoria, Onderstepoort, Pretoria, South Africa, 2 Department of Biomedical Sciences, Institute of Tropical Medicine, Antwerp, Belgium, 3 Faculty of Veterinary Medicine, Ghent University, Merelbeke, Belgium, 4 Section of Veterinary Public Health, Department of Paraclinical Sciences, Faculty of Veterinary Science, University of Pretoria, Pretoria, South Africa

* Nenene.Qekwana@up.ac.za

**Data Availability Statement:** All relevant data are within the manuscript and its Supporting information files.

## Abstract

Porcine cysticercosis and associated human infections are endemic in Sub-Saharan Africa, Latin America, and Asia. Poor agricultural practices, sanitary practices, and lack of knowledge increase the burden of the diseases in susceptible populations. This study investigates the seroprevalence of *Taenia* spp. in township pigs in Gauteng, South Africa and describes knowledge and farming practices of pig farmers regarding *T. solium* infections. Blood samples were collected from 126 pigs in three Gauteng township areas, and analyzed for active *Taenia* spp. infection using the B158/B60 Ag-ELISA. Farmer questionnaire surveys were conducted in four township areas to investigate the level of knowledge and practices associated with porcine cysticercosis and neurocysticercosis. Logistic regression models were used to assess the relationship between predictor variables and the outcome variable, knowledge of porcine cysticercosis or knowledge of neurocysticercosis. Overall, 7% of the pigs were seropositive for active *Taenia* spp. infection. 46% of farmers practiced a free-ranging system, while 25% practiced a semi-intensive system. Latrines were absent on all farms; however, 95% of farmers indicated that they have access to latrines at home. Most farmers had no knowledge of porcine cysticercosis (55%) or neurocysticercosis (79%), and this was not associated with any of the factors investigated. The prevalence of active *Taenia* spp. infection was reasonably low in this study, yet the knowledge level was also low, thus calling for further educational and training programmes to prevent *Taenia* spp. transmission in these communities.

## Introduction

*Taenia* species, *Taenia solium* and *Taenia hydatigena* have been reported as a cause of porcine cysticercosis in sub-Saharan Africa, Latin America, and Asia [1–4]. While *T. hydatigena* is not zoonotic, *T. solium* is the causative agent of human cysticercosis [5]. When the metacestode

**Funding:** NA Shongwe received funding from the Belgian Directorate-General for Development Co-operation Framework Agreement (FA4 DGDITM 2017-2021). URL:https://diplomatie.belgium.be/en The funders had no role in study design, data collection and analysis, decision to publish, or preparation of the manuscript.

**Competing interests:** The authors have declared that no competing interests exist.

larval form of the parasite develops in the brain of humans it results in neurocysticercosis, which may be associated with epilepsy, seizures and other neurological disorders [6, 7]. It is a major cause of acquired epilepsy in developing countries, commonly reported in children and older people [8–10].

Studies on *T. hydatigena* in pigs in Africa are limited. Nonetheless, there is evidence of the parasite circulating among the pig populations in Africa [4, 11]. Of concern is the cross reaction between *T. hydatigena* and *T. solium* in serological tests which hampers efforts to quantify the true prevalence of the two parasites in the pig population [12]. New research suggests that initial prevalences of *T. solium* in pig populations based on serological tests could have been overestimated [13]. Notwithstanding, porcine cysticercosis still poses a significant economic loss due to the reduced market value of infected carcasses [2, 14, 15].

Human cases of taeniasis and cysticercosis have been linked to the consumption of under-cooked or raw pork infested with *T. solium* cysticerci [16], poor hygiene and sanitation conditions and inadequate slaughtering facilities [17, 18]. Similarly, poor sanitary and hygiene conditions, and free-range systems have been associated with increased risk of *Taenia* spp. in pig populations [4, 19–21]. In addition, the level of education [22, 23], knowledge on livestock management [24, 25], and poor farming practices have been associated with high prevalence of *Taenia* spp. in pig populations [26, 27]. In contrast, mass drug administration in humans and pigs have been reported to reduce the risk of *T. solium* infections when implemented as an intregrated approach [25, 26, 28].

There is evidence of *Taenia* spp. circulating in pigs in Gauteng province, South Africa, with 14% pigs reported to be infected in selected areas [15]. In 2016, reports of illegally slaughtered pigs for human consumption and a high number of pigs with possible *T. solium* infection from areas under study surfaced. Furthermore, pig farmers in these areas were reported to have had poor husbandry practices and the farms were in close proximity to human settlements. In view of this, this study estimated the seroprevalence of active *Taenia* spp. in domestic pigs in selected areas of Gauteng and identified factors associated with infections and transmission based on knowledge and farming practices of pig owners. We hypothesize that *Taenia* spp. are circulating among pigs in township areas under study and that factors associated with the burden of *Taenia* spp. exist in these areas.

## Materials and methods

### Ethical considerations

Ethical clearance was obtained from the Animal Ethics Committee at the University of Pretoria, Faculty of Veterinary Science (Reference number: V070-18) as well as the Research Ethics Committee at the Faculty of Veterinary Science (REC 014–19). Further approval was obtained from the farmers before the commencement of this study. Signed informed consent was obtained from each respondent before the questionnaires were administered.

### Study area

Gauteng is a province situated in the Highveld of South Africa, occupying 1.4% of the land area of the country, and has an estimated population of 15,176,115 inhabitants [29]. Backyard farming, including pig farming, constitutes 89.5% of agricultural activities in the Gauteng Province [30]. It is estimated that 29.3% of adults in Gauteng live in poverty [31]. The province can be divided into three metropolitan cities, namely; Tshwane, Johannesburg, and Ekurhuleni [32].

The study focused on four township areas, one situated in Ekurhuleni, and three in Johannesburg. These areas were selected based on reports by veterinary extension services of poor

pig farming husbandry practices and proximity between farms and human settlements. In addition, there were reports of pigs in these areas being sold and illegally slaughtered for human consumption.

**Study population.** This study uses mixed methods to study the outlined objectives. Purposive sampling was used to identify all 56 farms and farmers to participate in this study. All pigs within the selected farms were tested provided that they met the following criteria i) Not pregnant, ii) older than 6 months, and iii) apparently healthy, therefore, making this step a probability sample [33]. The farmers were selected on the basis of known informal pig farming system, and known risk factors for *T. solium* [25]. A total of 56 farmers were approached through communication with the State Veterinarians and Animal Health Technicians (AHT) from the Department of Agriculture and Rural Development in Gauteng Province. The inclusion criteria for the farmers were as follows: i) must agree to be part of the study, ii) must be 18 years or older, and iii) must be involved in pig farming as an owner or employee. All (100%) of the identified farmers in each location agreed to be interviewed.

**Sample size calculation for seroprevalence.** The estimated sample size of 185 used to determine the seroprevalence of *Taenia* spp. in pigs was calculated using the following formula n:$\frac{z^2 p(1-p)}{\alpha^2}$ [33], where: z = 1.96, $p$ = expected prevalence of the disease, being 14% from a study done in Gauteng [15], and $\alpha = 0.05$ as the margin of error.

## Data collection

**Blood collection and processing.** One hundred and twenty-six blood samples were collected at three locations, in the Gauteng region. Blood samples were collected from the jugular vein of pigs into anticoagulant-free vacutainer tubes. The collected samples were stored in a polystyrene cooler box containing ice and transported to the Department of Veterinary Tropical Diseases laboratory at the Faculty of Veterinary Science, University of Pretoria. The blood was centrifuged, the serum was aliquoted into 2ml cryotubes and stored at -20˚C until analysis.

**Serological analysis.** The serum samples were tested for circulating *Taenia* spp. antigens using the B158/B60 Ag-ELISA commercial kit following the manufacturer's guidelines [34]. This assay does not differentiate between specific *Taenia* spp. detected in the serum. In Zambia, the in house version of the Ag-ELISA had a specificity (*Sp*) of 94.7% and sensitivity (*Se*) of 86.7% [35]. A study conducted in Zambia using the B158/B60 Ag-ELISA kit reported a specificity of 67% and sensitivity of 68% which increased to 90% in carcasses with one or more viable cysticerci and 100% for carcasses with more than 10 viable cysticerci, respectively [13]. While in the same study, the *Sp* of 49% and *Se* of 86% were reported in the identification of *T. hydatigena*.

A recently conducted study in Tanzania reported B158/B60 Ag-ELISA *Sp* and *Se* of 82.7% and 86.3%, respectively. However, the test *T. solium* positive predictive value was low (35.3%) compared to negative predictive values (98.2%) [36].

**Knowledge and practices among farmers.** A questionnaire was used to assess the knowledge and practices among pig farmers regarding porcine cysticercosis and neurocysticercosis. The questionnaire consisted of both closed and open-ended questions. Closed-type questions included checklists and selection type questions. While open-ended questions allowed for participants to elaborate on their opinions on the different topics in the questionnaire. The questionnaire was designed in Microsoft forms.

The questionnaire covers questions related to general knowledge and practices of farmers on *Taenia* spp. transmission and their attitudes towards neurocysticercosis-related illnesses. It was divided into the following themes: (i) demographic profile of respondents, (ii) knowledge on *Taenia* spp. infection and transmission, (iii) sanitary practices, and (iv) husbandry practices.

Prior to the data collection phase, the questionnaire was pre-tested by distribution to selected employees at the Faculty of Veterinary Sciences, officials at the Department of Agriculture and Rural Development, and a small group of individuals that were representative of the targeted population. The input obtained from the pre-testing was used to further modify the questionnaires.

Interviewers were researchers from the University of Pretoria, and they were trained on how to conduct interviews and found to be competent. The interviews were conducted in different local languages depending on the language spoken in that area or the language preferred by the farmer. The common languages spoken included IsiZulu, IsiXhosa, Sesotho, Afrikaans, and English. The average duration of the interviews was about 15 to 20 minutes.

## Data management and analysis

**General observation.** General observations about the environment at and around the smallholder pig farming were noted: proximity to households estimated in kilometers, sewage infrastructure, waste disposal, human access, and the presence of other animals.

**Descriptive statistics.** Seroprevalence results were managed using Microsoft excel. The proportions of positive samples and their 95% confidence interval were calculated based on location. Information from questionnaires was captured using Microsoft excel and checked for consistency and missing values.

Proportions of categoric variables and 95% confidence intervals were calculated and tabulated using the JASP software 11.1.0 (University of Amsterdam).

Thematic analysis as described by Braun and Clarke [37] was used to analyze the responses to the open-ended questions of the farmer questionnaire. The first phase was to become familiar with the questions in which the codes are to be created. Answers from these questions were then coded and where possible linked and grouped into potential themes. In addition, each theme was reviewed to assess whether it represents the underlying question. The final definitions and names of subthemes were then structured into the four main themes namely; knowledge of respondents, pig management, control practices and sanitary practices.

**Principal component analysis.** Principal Component Analysis (PCA) was used as a reduction tool to identify relationship among the different themes that were used as components in this method and the questions which are identified as factor loadings. Principal component analysis using an oblique rotation (Promax) was used to reduce a large set of possibly correlated variables into a smaller set for the logistic regression model. A parallel analysis was conducted to determine the number of components to be retained from the PCA [38]. The results of the parallel analysis were also visualized using a scree plot. The factor loading cut off value was set at 0.5 accounting for 25 percent of the variance of a variable. Five principal components with eigenvalues of above 1 were identified. The components were classified as follows; knowledge of respondents, deworming and sanitary practices, pig management, monitoring and control practices, and movement restriction.

**Logistic regression model.** Logistic regression models were fit to the data to assess the relationship between explanatory variables; town, level of education, source of income, the reason for farming, number of years farming, the number of pigs on the farm and the outcome variables knowledge of porcine cysticercosis (Yes/No) or knowledge of neurocysticercosis (Yes/No). In the first step, simple logistic models were fit between potential predictor variables and knowledge of porcine cysticercosis or neurocysticercosis. Variables that were significantly associated with the outcome at significance level, $\alpha = 0.20$ were considered for inclusion in the multivariable logistic regression model. In the multivariable logistic regression, the significance level of the predictor variables was set at $\alpha = 0.05$. Odds ratio (OR) and their 95% confidence intervals

(CI) were then computed for all variables in the univariable and multivariable models. The goodness-of-fitness of the model was assessed using the Hosmer-Lemeshow test statistic [39].

## Results

### General observations

All smallholder pig farms were in close proximity to households. However, the distance between farms and households differed by a township area: in area A, farms were on average 50m away from households, in area B, farms were 150m away from households, while in area C and area D, they were about 200m away. Smallholder farms were located downstream to the households and in some areas, sewage could be seen running along the streets and between houses towards the smallholder farms. Furthermore, there were dumping sites with used disposable diapers in sight located in close vicinity to the pig farming sites which were used for waste disposal. Children, livestock and dogs all had access to these dumping sites. The same dumping sites were used for the disposal of pig carcasses (Fig 1).

### Seroprevalence

Although the calculated samples size for this study was 185, only 126 pigs met the inclusion criteria outlined in the methods section. Seven percent (7%, 9/126) of pigs in this study were positive for *Taenia* spp circulating antigens. There was no significant association between the presence of *Taenia* spp. infection and township area (Table 1).

### Demographic profile of respondents

Almost 90% of the respondents in this study were male, and 52% had primary education or lower while 48% had secondary education or higher. Pig farming was the main source of income for the majority (95%) of respondents. Thirty percent (31%) of the respondents kept less than 10 pigs, while 48% of the respondents kept 11–20 pigs, and only 21% of respondents kept more than 20 pigs (Table 2).

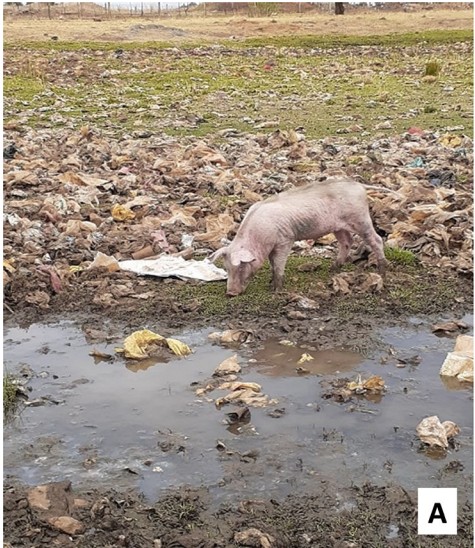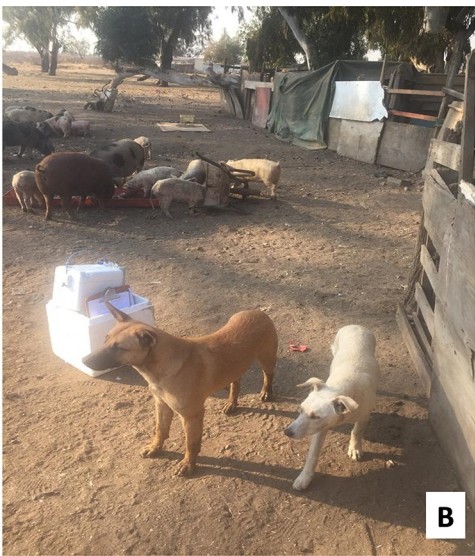

**Fig 1. Dumping sites around pig farms in township areas in Ekurhuleni District, Gauteng Province, South Africa.** (A) A pig feeding around sewage; (B) Dogs and pigs roaming or feeding around dumping sites.

**Table 1. Seroprevalence of *Taenia* spp., as determined by the Ag-ELISA, in pigs from farms in Gauteng Province.**

| Area | Animals tested | | Animals positive for *Taenia* species | | | |
|------|--------|------------|----------|------------|----------|-------|
| | Tested | Proportion | Positive | Percentage | 95% CI[a] | |
| D | 72 | 57.1 | 6 | 8.3 | 3.88 | 17.01 |
| C | 36 | 28.6 | 2 | 5.6 | 1.54 | 18.14 |
| B | 18 | 14.3 | 1 | 5.6 | 0.99 | 25.76 |
| Total | | | | | | |

[a]95% Confidence interval

**Farming practice and sanitation among farmers.** Forty-six percent (46%) of farmers indicated that they practiced a free-ranging system while 25% mentioned they practiced a semi-intensive system. The majority (77%) of all the farmers indicated that their animals were born on the farm. Most (86%) farmers did not introduce new stock on their herds. Of those that introduced new stock (14%), they indicated that they sourced the animals from neighboring farmers (77%) and auctions (23%). Most pigs (75%) were fed mainly on kitchen waste mixed with commercial feed (Table 2).

Respondents (68%) indicated they dewormed their pigs more than once a year (74%). Out of those that dewormed pigs, 66% used ivermectin, while 26% did not know the type of drug they used for deworming, and on the other hand, 5% used antibiotics for deworming. Most farmers (65%) indicated that they deworm all the pigs while 18% deworm only piglets or adults (Table 2). For farmers that did not deworm their pigs, the reasons given included the cost of treatment and dependency on the government for treatments.

None of the areas visited had latrines available around the farms. However, the majority (95%) of farmers indicated that they have access to latrines at home and always used them (89%). Farmers that occasionally used latrines (11%) indicated that they additionally used bushes for defecation or the open fields. Most (93%) respondents used plain water to wash their hands after using the latrine or defecating in the bushes. Only 5% of the respondents used both soap and water to wash hands after using the latrine. Two respondent did not wash hands after using the latrine (Table 2).

**Knowledge of porcine cysticercosis and neurocysticercosis among farmers.** Fifty-five percent (55%) of the respondents indicated that they have no knowledge of cysticercosis in pigs, and 79% had never heard about neurocysticercosis. Those who had knowledge about cysticercosis (45%) indicated that they obtained it through agricultural workshops, auctions, and media. Sixteen (16%) percent of respondents knew both how the pig acquired the disease and how to detect cysts on pigs. However, only 14% knew where to find the cyst in a pig (Table 3).

The majority (68%) of the respondents indicated that they sold live pigs only and only 2% slaughtered pigs solely for their own consumption. Fifty-seven (57%) of farmers indicated that they slaughtered pigs at home for own consumption or selling yet only 30% performed meat inspection. When asked what they looked for during meat inspection, the farmers mentioned white nodules on the offal, carcass, liver, trachea, lung, and tongue. Some farmers mentioned that they looked for pleuritis, discoloration of offal and liver, cysts in the muscle and milk spots on the liver. Most (88%) respondents indicated that if an abnormality is found in the carcass, the meat is discarded by either burning the carcass and burying or feeding the carcass to the dogs. One person indicated that they sell the meat as it is and three respondents indicated they consumed the meat anyway with one saying (Table 3).

"*meat is meat, it cannot be thrown away*".

**Table 2. Demographic profile and farming practices of smallholder farmers in four township areas in Gauteng (n = 56).**

| Variable | Category | Percent |
|---|---|---|
| **Gender** | Female | 11 |
| | Male | 89 |
| **Age group (years)** | 25–40 | 25 |
| | 41–60 | 46 |
| | >60 | 29 |
| **Education** | ≤Primary | 52 |
| | ≥Secondary | 48 |
| **Source of income** | Farming | 94 |
| | Employed | 4 |
| | Pensioner | 2 |
| **Number of pigs kept** | <10 | 31 |
| | 11–20 | 48 |
| | >20 | 21 |
| **Production system** | Free-range | 47 |
| | Intensive | 29 |
| | Semi-intensive | 25 |
| **Source of current pig stock** | Born on farm | 77 |
| | Auction | 23 |
| **Introduction of new pigs** | No | 86 |
| | Yes | 14 |
| **Pig feed source** | Kitchen waste | 23 |
| | Commercial feed | 2 |
| | Kitchen waste and commercial feed | 75 |
| **Deworming of pigs** | Yes | 68 |
| | No | 32 |
| **Frequency of deworming** | Once a year | 26 |
| | More than once a year | 74 |
| **Treatment coverage** | All Pigs | 63 |
| | Adult pigs | 18 |
| | Piglets | 18 |
| **Name of dewormer** | Don't know name | 26 |
| | Ivermectin | 66 |
| | Tetracycline | 8 |
| **Access to latrine** | Yes | 95 |
| | No | 6 |
| **Frequency of usage** | Always | 89 |
| | Sometimes | 11 |
| **Hand washing method** | Water only | 93 |
| | Both water and soap | 5 |
| | None | 2 |

In the univariate model, only the level of education of the respondents and pig farming purpose were considered for the multivariate model.

In the multivariable model, there was no significant association between level of education ($p = 0.0929$), pig farming purpose ($p = 0.1286$) and outcome, knowledge of neurocysticercosis (Table 4).

**Table 3. Questions relating to knowledge porcine cysticercosis and neurocysticercosis among farmers in four township areas in Gauteng (n = 56).**

| Variable | Category | Number | Percent |
|---|---|---|---|
| **Heard about porcine cysticercosis** | Yes | 25 | 45 |
| | No | 31 | 55 |
| **If Yes from where** | Workshop | 8 | 32 |
| | Auction | 12 | 48 |
| | Media | 5 | 20 |
| **Heard about neurocysticercosis** | Yes | 12 | 21 |
| | No | 44 | 79 |
| **Have you received training on *Taenia* species** | Yes | 12 | 21 |
| | No | 44 | 79 |
| **Do you know how pigs acquire *T. solium*** | Yes | 9 | 16 |
| | No | 47 | 84 |
| **Do you know where to find cysts in a pig** | Yes | 8 | 14 |
| | No | 48 | 86 |
| **Do you know how to detect cysts in a pig** | Yes | 9 | 16 |
| | No | 47 | 84 |
| **Farming purpose** | Selling live pigs only | 38 | 68 |
| | Own consumption | 1 | 2 |
| | Selling pork meat | 6 | 11 |
| | All three | 11 | 20 |
| **Slaughter pigs at home** | Yes | 32 | 57 |
| | No | 24 | 43 |
| **Do you perform meat inspection** | Yes | 17 | 30 |
| | No | 39 | 70 |
| **What do you look for when inspecting** | White nodules | 10 | 59 |
| | Pleuritis | 2 | 12 |
| | Discolouration | 2 | 12 |
| | Cyst | 2 | 12 |
| | Milk spots | 1 | 6 |
| **Action when abnormalities are found** | Discard | 28 | 88 |
| | Sell | 1 | 3 |
| | Consume | 3 | 9 |

The results of the univariable model shows no significant association between town, level of education ($p = 0.6109$), source of income ($p = 0.5395$), pig farming purpose ($p = 0.2611$), years in pig farming ($p = 0.4600$), pig numbers kept by farmers ($p = 0.8240$) and outcome, knowledge of porcine cysticercosis (Table 5).

## Discussion

The objectives of this study were to investigate the seroprevalence of *Taenia* species in pigs as well as knowledge and practices associated with porcine cysticercosis and neurocysticercosis among smallholder pig farmers in Gauteng.

### Seroprevalence of *Taenia* spp.

We observed a lower (7%) proportion of pigs positive for *Taenia* spp. infections in this study compared to the 14%, 34%, and 54.8% reported in Gauteng, Free State [15], and Eastern Cape [40] areas, respectively.

**Table 4. Predictors of knowledge of neurocysticercosis.**

| Predictors | Univariable model | | | | Multivariable Model | | | |
|---|---|---|---|---|---|---|---|---|
| | OR[a] | 95% CI[b] | | p-value | OR[a] | 95% CI[b] | | p-value |
| Town[&] | | | | 0.6509 | | | | |
| D | 0.79 | 0.18 | 3.39 | 0.6631 | | | | |
| C | 0.32 | 0.03 | 3.56 | 0.3697 | | | | |
| B | Ref | - | - | - | | | | |
| Education | | | | | | | | |
| None-Primary | 0.38 | 0.10 | 1.45 | 0.1571 | 0.30 | 0.08 | 1.22 | 0.0929 |
| Secondary-Tertiary | Ref | - | - | - | Ref | - | - | - |
| Source of Income | | | | | | | | |
| Farming | 0.67 | 0.15 | 3.03 | 0.5996 | | | | |
| Farming and additional jobs | Ref | - | - | - | | | | |
| Pig farming purpose | | | | | | | | |
| Selling live pigs | 2.86 | 0.56 | 14.70 | 0.2090 | 3.74 | 0.68 | 20.49 | 0.1286 |
| All others | Ref | - | - | - | Ref | - | - | - |
| Years in pig farming | 1.10 | 0.93 | 1.31 | 0.2620 | | | | |
| Pig numbers | 1.00 | 0.99 | 1.02 | 0.6676 | | | | |

[&]: A was removed due to all participants responding "no" to this question

[a]Odds ratio

[b]95% Confidence interval

The presence of pigs infected with *Taenia* spp. can be attributed to poor sanitary conditions such as sewage spillage and the presence of open disposal of waste sites accessible to the pigs observed in this study. Moreover, farmers (71%) practiced a free-roaming pig system which has been previously reported as a risk factor for the transmission of *Taenia*

**Table 5. Univariable model for predictors of knowledge of porcine cysticercosis.**

| Predictors | Univariable model | | | |
|---|---|---|---|---|
| | OR[a] | 95% CI[b] | | p-value |
| Town | | | | 0.5381 |
| A | 0.39 | 0.06 | 2.70 | |
| D | 0.93 | 0.25 | 3.52 | |
| C | 1.94 | 0.32 | 11.76 | |
| B | Ref | - | - | - |
| Education | | | | |
| None or primary | 0.76 | 0.26 | 2.19 | 0.6109 |
| Secondary or tertiary | Ref | - | - | - |
| Source of Income | | | | |
| Farming | 0.65 | 0.17 | 2.55 | 0.5395 |
| Farming and additional jobs | Ref | - | - | - |
| Pig farming Purpose | | | | |
| Selling live pigs | 0.52 | 0.17 | 1.62 | 0.2611 |
| All others | Ref | - | - | - |
| Years in pig farming | 1.06 | 0.91 | 1.23 | 0.4600 |
| Pig numbers | 1.00 | 0.98 | 1.02 | 0.8240 |

[a]Odds ratio

[b]95% Confidence interval

spp infections [25, 41–43]. In addition, since dogs and pigs coexist in the same environment, the pigs could be infected with either *T. solium* or *T. hydatigena* or both *Taenia* spp. Although the results of this study indicate the presence of *Taenia* spp. among pigs in these communities, further diagnosis using carcass dissection and molecular confirmation using PCR is required to confirm infection with either *T. solium* or *T. hydatigena* or both since the Ag-ELISA used is genus-characteristic [44, 45].

The proportion of infected pigs in this study did not differ between the three townships tested. Similarly, Tsotetsi et al. [15] in South Africa and Sikasunge et al [46] reported no regional significant difference in the prevalence of porcine cysticercosis. In contrast, variations in the prevalence of *Taenia* spp. infections have been reported among villages in Tanzania [42].

The results suggest that factors influencing the epidemiology of *Taenia* spp. in pigs from smallholder farms in Gauteng are similar across the study locations. In view of this, mitigation strategies such as confinement of pigs to limit access to the dumping sites and management of contaminated water sources must be applied to all the smallholder farms in the studied township areas [29].

## Demographic profiles and farming practices

Th majority (75%) of the farmers in this study were above 40 years old. Myeni and others [47] reported similar findings in the Free-State province smallholder farming communities. It was surprising to see that almost half (48%) of farmers had secondary education or higher contrary to the findings of other studies which reported a lower level of education among smallholder farmers in South Africa [31, 47, 48]. The characteristic profile of the population in this study suggests that risk communication and control strategies for *Taenia* spp. infection and transmission must be targeted towards middle aged males. In addition, it is likely that this targeted population will be under-resourced and unable to afford expensive control programs as observed in this study.

Most farmers indicated that they deworm their pigs at least once a year using ivermectin. Similarly, farmers in Zambia and Tanzania routinely used ivermectin to deworm their pigs [24]. The proportion of farmers using unknown medicine and antibiotics as regular deworming drugs is alarming and an indication of the poor knowledge level among farmers.

Although this suggests that farmers are aware of the importance of animal health, the anti-parasitic drugs used are not effective against *Taenia* spp. infections. In view of this, farmers should be educated on the treatment coverage of anti-parasitic drugs with emphasis on the usage of drugs reported to be effective in *Taenia* spp. treatment such as, oxfendazole. Farmers should also be discouraged from buying animals from sources such as auctions, where the health status of pigs is often unknown.

## Risk factors for *Taenia* spp. transmission

Few (6%) farmers in this study did not have access to toilets, which is comparable with the 16% reported among rural farmers in a previous study in Gauteng Province [25]. Although this is a positive outcome, access to latrines does not guarantee their use [28, 40, 42, 49]. Nonetheless, the results of this study suggest that access to latrine may not be a major risk factor for the occurrence of cysticercosis in Gauteng since most farmers had access to latrines. Notwithstanding, farmers who defecate on the open field must be made aware that this practice is likely to increase the risk of environmental contamination and the prevalence of *T. solium* among pigs in the area [15, 25, 28].

Informal slaughter of pigs was common (57%) among farmers in this study, and the majority of the slaughter was performed without meat inspection (70%). Similarly, a previous study

in Gauteng reported that 85% of smallholder farmers did not slaughter their livestock in a registered abattoir [25]. Although the sensitivity of meat inspection for porcine cysticercosis is very low [35], the need for meat inspection as a control measure cannot be overemphasized as availability and adequacy of meat inspection services are important mitigation steps for *Taenia* spp. infection and neurocysticercosis [26, 50, 51]. Moreover, farmers in this study indicated that when performing inspection, they identified white nodules, pleuritis, and cysts.

In addition, the consumption of infected pork as reported by few (11%) farmers and feeding of infected meat to the dogs must be discouraged as this also may facilitate transmission of *T. hydatigena* [4, 52, 53]. Furthermore, children, dogs, and pigs shared the same environment in this study. Therefore, it is possible that this interface plays a significant role in the epidemiology of *Taenia* spp. infections in this environment [40, 54, 55]. Therefore, further studies are needed to investigate the burden of neurocysticercosis among children and consumers of pork meat sources from these areas. In addition, communities must be educated on measures that they can implement to minimize the risk of *Taenia* spp. infection. Farmers and consumers must be educated on the implications of selling, purchasing and consuming possibly infected uninspected meat.

## Knowledge on porcine cysticercosis and neurocysticercosis

Almost half (45%) of the respondents had heard about porcine cysticercosis compared to those that were aware of neurocysticercosis which was similarly reported in Uganda [48] and Tanzania [56]. In addition, almost all farmers interviewed did not know how the pig acquires the infection. This may be attributed to a lack of education on the epidemiology of *Taenia* spp. observed in this study which has been previously reported as a risk factor for *T. solium* transmission [48]. These results are not surprising as lack of awareness on the epidemiology of *T. solium* has previously been reported in other African countries with 0.6% in Burkina Faso [26] and 28.6% in Cameroon [27]. Moreover, respondents in this study could not link the identified postmortem abnormalities with the potential health risks involving *Taenia* spp. In view of this, the use of media, workshops, and veterinary extension is encouraged as improved knowledge has been linked to reduced levels of *T. solium* infections [57, 58]. These knowledge transfer outreach strategies should not be targeted towards a specific group, age, gender or practice since these were not significant factors influencing the level of knowledge on porcine cysticercosis or neurocysticercosis in this study.

This study is not without limitations, the reported African Swine Fever outbreak in between April and July 2019 restricted movement between areas hence pigs from township area A could not be sampled. In addition, pig farmers were unwilling to sell their pigs therefore pig carcasses or offal could not be purchased for further analysis including carcass dissection and PCR. The Ag-ELISA used in this study was not species-specific therefore we could not differentiate infections with *T. hydatigena* and *T. solium*.

## Conclusions

The *Taenia* spp. seroprevalence in this study was low compared to that from other studies conducted in South Africa and other African countries. Knowledge level of farmers regarding *Taenia* spp. in this study was low and did not differ based on education, level, and years in practice. Factors previously associated with the epidemiology of taeniasis and neurocysticercosis including free-roaming of pigs, lack of meat inspection, lack of knowledge of the disease and sewage spillage were identified in this study. Therefore, there is a need for education and training of farmers on the epidemiology of porcine and human cysticercosis to mitigate the risk of *Taenia* spp. infections among farmers and consumers in Gauteng township areas.

## Supporting information

**S1 Appendix. Consent form for farmers to participate in the study.**
(PDF)

**S2 Appendix.**
(PDF)

## Acknowledgments

The authors would like to acknowledge the Gauteng Department of Agriculture and Rural Development, the Institute of Tropical Medicine, the National Research Foundation and the University of Pretoria for their contributions in this study. We would also like to thank all participating farmers for having their pigs sampled and for their willingness to respond to the questionnaires.

## Author Contributions

**Conceptualization:** Nothando Altrecia Shongwe, Pierre Dorny, Veronique Dermauw, Daniel Nenene Qekwana.

**Data curation:** Nothando Altrecia Shongwe, Charles Byaruhanga.

**Formal analysis:** Daniel Nenene Qekwana.

**Funding acquisition:** Pierre Dorny, Veronique Dermauw.

**Investigation:** Nothando Altrecia Shongwe.

**Methodology:** Nothando Altrecia Shongwe, Charles Byaruhanga, Pierre Dorny, Veronique Dermauw.

**Project administration:** Nothando Altrecia Shongwe, Daniel Nenene Qekwana.

**Resources:** Pierre Dorny, Veronique Dermauw, Daniel Nenene Qekwana.

**Supervision:** Pierre Dorny, Veronique Dermauw, Daniel Nenene Qekwana.

**Validation:** Pierre Dorny, Daniel Nenene Qekwana.

**Writing – original draft:** Nothando Altrecia Shongwe.

**Writing – review & editing:** Nothando Altrecia Shongwe, Charles Byaruhanga, Pierre Dorny, Veronique Dermauw, Daniel Nenene Qekwana.

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
