## [Decision Letter · Decision Letter 0]

29 May 2020

PONE-D-20-14308

Taenia species in smallholder farms in Gauteng, South Africa: seroprevalence in pigs, and farmers’ knowledge and practices

PLOS ONE

Dear Dr. Daniel Nenene Qekwana

Thank you for submitting your manuscript to PLOS ONE. After careful consideration, we feel that it has merit but does not fully meet PLOS ONE’s publication criteria as it currently stands. Therefore, we invite you to submit a revised version of the manuscript that addresses the points raised during the review process.

Many thanks for submitting your manuscript to PLOS One

Your manuscript was reviewed by three experts in the field, and they have recommended some modifications be made prior to acceptance

If you could write a detailed response to reviewers, that will help to expedite review when resubmitted.

I wish you the best of luck with your revisions

Hope you are keeping safe and well in these difficult times

Many thanks

Simon

We look forward to receiving your revised manuscript.

Kind regards,

Simon Clegg, PhD

Academic Editor

PLOS ONE

2. Please include additional information regarding the survey or questionnaire used in the study and ensure that you have provided sufficient details that others could replicate the analyses. For instance, please describe the pretesting of this tool in further detail, including the number of participants and whether or not they provided written informed consent before consultation.

3.Please amend either the title on the online submission form (via Edit Submission) or the title in the manuscript so that they are identical.

4. Your ethics statement must appear in the Methods section of your manuscript. If your ethics statement is written in any section besides the Methods, please move it to the Methods section and delete it from any other section. Please also ensure that your ethics statement is included in your manuscript, as the ethics section of your online submission will not be published alongside your manuscript.

(Note: HTML markup is below. Please do not edit)

Reviewers' comments:

Reviewer's Responses to Questions

**Comments to the Author**

1. Is the manuscript technically sound, and do the data support the conclusions?

Reviewer #1: Partly

Reviewer #2: Yes

Reviewer #3: Yes

2. Has the statistical analysis been performed appropriately and rigorously? 

Reviewer #1: I Don't Know

Reviewer #2: Yes

Reviewer #3: Yes

3. Have the authors made all data underlying the findings in their manuscript fully available?

Reviewer #1: Yes

Reviewer #2: Yes

Reviewer #3: Yes

4. Is the manuscript presented in an intelligible fashion and written in standard English?

Reviewer #1: Yes

Reviewer #2: No

Reviewer #3: Yes

5. Review Comments to the Author

Reviewer #1: This manuscript describes a study which undertook serology on pigs living in a region of South Africa as well as a survey of farmers knowledge and attitudes related to the transmission of Taenia solium. While the manuscript’s title and abstract correctly refer to the serological outcomes as providing evidence of infection with Taenia spp, rather than T. solium specifically, all other aspects of the paper are presented as if the work was relevant specifically to T. solium. Unfortunately, the study did not obtain any direct evidence that any of the pigs involved actually had T. solium infection. The serological test that was used in the study is unable to differentiate infection with T. solium from infection with T. hydatigena. Infection with T. hydatigena has no relevance as a zoonotic infection nor would it have any relevance to the survey that was conducted. The value of the study would have been increased enormously had the Ag-ELISA positive pigs been purchased and Taenia infection, or more importantly, T. solium infection been confirmed at necropsy.

The limitations to the interpretation of the serological data in relation to T. solium place a major constraint on the importance of the study. Nevertheless, the results are worthy of publication, but the many interpretations and analyses relating to T. solium are not warranted. The valuable data about both seroprevalence and farming practices are publishable as statements of fact, without the extensive analyses that assume the data relates to T. solium when this has not been determined. Hence my recommendation is that the manuscript be re-drafted and shortened substantially.

The following are some notes to be considered if a revised manuscript is prepared.

Line 101. Add full stop.

Lines 111-114. Reference 33 does not provide the methodology that was used for the commercial Ag-ELISA. The commercially available test uses a methodology significantly different to the in-house test developed by the Institute for Tropical Medicine in Belgium which was used in the work described in Reference 33. The test’s specificity and sensitivity, referred to in Reference 11, uses the in-house test methodology and not the commercial kit. To my knowledge there has been no proper evaluation of the sensitivity and specificity of the commercial kit. There have been numerous reports of the sensitivity and specificity of the B158/B60 Ag-ELISA, several of which have found the test to have a substantially lower specificity for detection of T. solium infection than the single reference chosen to be cited here. The choice of this single reference gives a biased impression of the test’s performance.

Lines 284-6. The authors cannot attribute the prevalence of pigs infected with Taenia spp to the conditions they describe because they have no idea whether any were actually infected with T. solium. It is possible that all the positives were infected with T. hydatigena or, given the occurrence of false positive pigs in the Ag-ELISA, they may be pigs with no Taenia infection at all. The authors should qualify these interpretations by stating “If the Ag-ELISA positive pigs were infected with T. solium….” and include a statement indicating that the test positive animals may instead harbour T. hydatigena, in which case it would be an indication of the presence of dogs in the community which were not regularly treated with a taeniacide which had access to offal from pigs, sheep or goats.

Lines 288-291. It is incorrect to say that they would need carcass dissection plus PCR to differentiate animals infected with T. solium and T. hydatigena. For the vast majority of infections with either of these two parasites the identity of the species involved is clear based on the cysticercus morphology and site of encystment. It is recommended that the words ‘coupled with PCR’ be deleted.

Lines 296-298. Care needs to be exercised in commenting on the results of other publications which involved use of serology for determining the prevalence of T. solium. For may years the Ag-ELISA test was used and the fact that the test is not species specific was virtually ignored and the results assumed to relate to T. solium. Recent detailed evaluations of the test’s performance have revealed that there is a high proportion of animals that are positive in the test that have neither T. solium nor T. hydatigena infected. For example, Chembensofu et al 2017 (DOI 10.1186/s13071-017-2520-y) found that from 30 pig carcasses that were fully dissected and found negative for T. solium, the Ag-ELISA returned ten positives, of which two were infected with T. hydatigena. That is 8/30 (26.6%) animals that had neither T. solium nor T. hydatigena infection tested false positive in Ag-ELISA. With a false positive rate of this magnitude, it is not inconceivable that all 9 Ag-ELISA positive animals from the 126 that were tested in the study described in this manuscript may have no Taenia infection at all…

Lines 352-353. The sharing of the environment of children and pigs with dogs has nothing whatsoever to do with transmission of T. solium.

Lines 400-401. It is difficult to understand which authors did what – it appears that the letters refer to authors’ Christian names. It is suggested to use all the authors’ initials.

Reviewer #2: MANUSCRIPT REVIEW REPORT

Dear Editor (PLOS ONE)

Kindly find a REVIEW REPORT for a manuscript titled “Taenia spp. in smallholder farms in Gauteng, South Africa: seroprevalence in pigs, and farmers’ knowledge and practices”. The report contains general comments as well as specific comments for the different sections.

GENERAL COMMENTS

The authors of this manuscript are reporting findings on the epidemiology of an important zoonotic parasite of pigs. Somewhere, in the introduction (lines 64 – 66) and in the discussion (lines 280-283) the authors clearly indicate that similar studies have been done in the same area. In my opinion, results of an undertaking reported here are just a duplication of something which has already been reported and doesn’t add any value. It is however upon the journal Editorial Board to decide on the fate of this manuscript.

Technically, the article is fairly well written although there are some minor language problems and some study design issues which are raised in section reviews.

COMMENTS BY SECTION

TITLE

The authors need to recast the title, they could write it better than the way it is currently

ABSTRACT

Line 15-16: The sentence “Poor agricultural practices, and sanitary practices, and lack of…………” should be re-written to reduce the number of “ands”

Line 22-25: The sentence “Descriptive statistics were run and logistic regression models were used to assess the relationship between socio demographic and pig farming conditions and knowledge of porcine cysticercosis and neurocysticercosis.” should be re-written to reduce the number of “ands”

INTRODUCTION

Lines 64 – 66: the authors could not clearly indicate the rationale of conducting the study

MATERIALS AND METHODS

In line 86 the authors indicate the use of mixed method which are not evident in the subsequent part of the study design.

In line 86-87 the authors state that they used purposive sampling to identify farms and farmers to participate in this study, they should clearly explain/reason justification for this.

Line 87-90: the authors explain the selection procedure of pigs to be involved in the study and they eventually refer to the step as being probability, THIS IS SERIOUS.

Line 87-88: the authors indicate that all pigs that met criteria in all the selected farms were sampled, what was the essence of calculating the sample size then?

The authors also need to state the number of farms from which pigs were sampled;

Line 90-95: the authors explain the selection procedure of farmers. It is not clear whether these were the owners of the farms where pigs were sampled or otherwise.

Line 98: the authors indicate that the calculated sample size was 185 pigs, but in Line 101: they indicate they sampled 126 pigs, this needs explanation.

Lines 103-114: The authors do not consider “Blood collection and processing” and “Serological analysis” as part of data collection, can they give a reason to this?

Lines 128-131: The pretesting of the questionnaire doesn’t seem to make sense as the population to which it was tested is very different in characteristics from farmers. The authors need to defend their decision.

Line 150: The sentence “The open-ended questions were analyzed using thematic analysis” should be cancelled.

RESULTS

Generally this section is poorly organized, the flow is not good

In several occasions there is double presentation of results in text and in table

DISCUSSION

Generally the discussion is poorly organized

References “7” cited in line 280 and reference “36” cited in line 283 indicate that similar works were conducted in the same area. This is a weakness as resources devoted in this study could be deployed elsewhere

CONCLUSIONS

The third word in line 385 (training) does sound good to me as it is linked to imparting farmers knowledge on epidemiology of the parasite

Ethical considerations

Line 389-390: Check this “The Animal Ethics Committee Research Ethics Committee”

Line 393-394: I am doubting on the statement that; “Signed informed consent was obtained from each respondent before the questionnaires were administered”. This implies that all farmers could read and write

Acknowledgements

Line 396: The statement “The author would like to acknowledge the Gauteng……….” Indicates that this manuscript is single authored

Reviewer #3: The manuscript is interesting and very well written. I have a few comments:

All the manuscript always refers to Taenia spp, but since all the study was performed in pigs and since it also states human neurocysticercosis, I recommend to change spp for solium, mainly because it is more accurate even though the species was not identified but Taenia saginata is only found in cattle and it does not cause neurocysticercosis in humans and using T. soliun more precise in educational terms. An exception is lines 290-291

The authors should include the concept of KAP because they use it but it is not included as a specific issue such as in lines 117, 127, 157.

The reference to the statistic test cited on lines 182-183 should be provided.

Line 210 does not state pigs in numbers between 10 and 20. The authors should add: 11-20 pigs, 48%

Line 228, if not plain water, what other source/type of water did the farmers use?

Table 3. farming purpose “all three” should be moved two lines below

Line 310 eliminate “of age”

The subject of KAP should also be included in the paragraph that starts on line 359

In acknowledgments I would add to the farmers that participated in the study

6. PLOS authors have the option to publish the peer review history of their article (what does this mean?). If published, this will include your full peer review and any attached files.

Reviewer #1: No

---

## [Author Response · Author response to Decision Letter 0]

20 Aug 2020

Reviewer 1

Comments: While the manuscript’s title and abstract correctly refer to the serological outcomes as providing evidence of infection with Taenia spp, rather than T. solium specifically, all other aspects of the paper are presented as if the work was relevant specifically to T. solium. Unfortunately, the study did not obtain any direct evidence that any of the pigs involved actually had T. solium infection. The serological test that was used in the study is unable to differentiate infection with T. solium from infection with T. hydatigena.

Responses: We thank the reviewer for the comments, and we have subsequently amended the introduction section of the manuscript in line with the suggestions made. Please also see our responses below.

Comments: Infection with T. hydatigena has no relevance as a zoonotic infection nor would it have any relevance to the survey that was conducted. The value of the study would have been increased enormously had the Ag-ELISA positive pigs been purchased and Taenia infection, or more importantly, T. solium infection been confirmed at necropsy.

Response: We agree with the reviewer that purchasing of positive cases would have contributed significantly to the finding of this study. We highlighted this aspect as one of the limiting factors in the discussion section of the manuscript. We have also indicated that we could not purchase any animals due to movement restriction as results of the African Swine Fever outbreak. 

Although we agree that the necropsy could have assisted in the identification T. solium, it is important to note that this method has been shown to underestimate the presence of T. solium in carcasses with very few cysts: 

Gavidia, C.M., Verastegui, M.R., Garcia, H.H., Lopez-Urbina, T., Tsang, V.C., Pan, W., Gilman, R.H., Gonzalez, A.E. and Cysticercosis Working Group in Peru, 2013. Relationship between serum antibodies and Taenia solium larvae burden in pigs raised in field conditions. PLoS Negl Trop Dis, 7(5), p.e2192.

Lightowlers, M.W., Assana, E., Jayashi, C.M., Gauci, C.G. and Donadeu, M., 2015. Sensitivity of partial carcass dissection for assessment of porcine cysticercosis at necropsy. International journal for parasitology, 45(13), pp.815-818.

Comments: The many interpretations and analyses relating to T. solium are not warranted. 

Response: The reviewer’s comments have been noted. However, it is important to highlight that the focus of the study was to investigate the prevalence of Taenia spp. and assess the level of knowledge of taeniasis and cysticercosis in relation to T. solium specifically.

Comments: The valuable data about both seroprevalence and farming practices are publishable as statements of fact, without the extensive analyses that assume the data relates to T. solium when this has not been determined. Hence my recommendation is that the manuscript be re-drafted and shortened substantially.

Response: We thank the reviewer for the comment. The analysis relates to factors associated with knowledge of neurocysticercosis and porcine cysticercosis. Please see the objectives of the study.

Comments: Lines 111-114. Reference 33 does not provide the methodology that was used for the commercial Ag-ELISA. The commercially available test uses a methodology significantly different to the in-house test developed by the Institute for Tropical Medicine in Belgium which was used in the work described in Reference 33. The test’s specificity and sensitivity, referred to in Reference 11, uses the in-house test methodology and not the commercial kit. To my knowledge there has been no proper evaluation of the sensitivity and specificity of the commercial kit. There have been numerous reports of the sensitivity and specificity of the B158/B60 Ag-ELISA, several of which have found the test to have a substantially lower specificity for detection of T. solium infection than the single reference chosen to be cited here. The choice of this single reference gives a biased impression of the test’s performance.

Response: This has been noted. The supporting literature for the different specificity and sensitivity values of the test have been provided. Although the commercial test has not been systemically validated, and the literature on the B158/B60 Ag-ELISA pertains to the in house test, the commercial test differs from the in house test only by a shorter procedure, a different substrate and a different method of calculating the cut off. The monoclonal antibodies in both assays are similar. The company commercialising the Ag-ELISA has done a validation of their kit against the in-house test. Updated comparative results of the B158/B60 Ag-ELISA against validated methods have been provided by Chembensofu et al., 2017. In their study, the authors show that sensitivity of the Ag-ELISA to detect carcasses with viable cysticerci (1 or more) was estimated at 91% (95% CI: 71–99%), and increased further to 100% (95% CI: 75–100%) for carcasses with at least 10 viable cysticerci. The specificity of the Ag-ELISA to detect infected carcasses was estimated at 67%.

Comments: Lines 284-6. The authors cannot attribute the prevalence of pigs infected with Taenia spp to the conditions they describe because they have no idea whether any were actually infected with T. solium. It is possible that all the positives were infected with T. hydatigena or, given the occurrence of false positive pigs in the Ag-ELISA, they may be pigs with no Taenia infection at all. The authors should qualify these interpretations by stating “If the Ag-ELISA positive pigs were infected with T. solium….” and include a statement indicating that the test positive animals may instead harbour T. hydatigena, in which case it would be an indication of the presence of dogs in the community which were not regularly treated with a taeniacide which had access to offal from pigs, sheep or goats.

Response: The comments have been noted. The sentence has been revised as “In addition, since dogs and pigs coexist in the same environment, the pigs could be infected with either T. solium or T. hydatigena or both Taenia spp.”

 Taenia spp. comment, there is sufficient literature to support the assumption that the existing predisposing factors in these areas contribute to the presence of Taenia spp. in this study. Taenia hydatigena similar to T. solium have been reported in conditions where pigs are free-roaming and scavenging with increased exposure to contaminated pastures, feeds, and water. In addition, poor management practices have been reported to facilitate T. hydatigena transmission in areas where dogs or wild animals share the same environment. There are also studies that have reported co-infection between T. hydatigena and T. solium in areas where pigs share the same environmental conditions:

Braae, U.C., Kabululu, M., Nørmark, M.E., Nejsum, P., Ngowi, H.A. & Johansen, M.V. 2015. Taenia hydatigena cysticercosis in slaughtered pigs, goats, and sheep in Tanzania. Tropical Animal Health and Production. 47(8):1523–1530.

Chembensofu, M., Mwape, K.E., Van Damme, I., Hobbs, E., Phiri, I.K., Masuku, M., Zulu, G., Colston, A., et al. 2017. Re-visiting the detection of porcine cysticercosis based on full carcass dissections of naturally Taenia solium infected pigs. Parasites and Vectors. 10(1):1–9.

Delano, M.L., Mischler, S.A. & Underwood, W.J. 2002. Biology and Diseases of Ruminants : Second Edition ed. Elsevier Inc.

Dermauw, V., Ganaba, R., Cissé, A., Ouedraogo, B., Millogo, A., Tarnagda, Z., Hul, A. Van, Gabriël, S., et al. 2016. Taenia hydatigena in pigs in Burkina Faso: A cross-sectional abattoir study. Veterinary Parasitology. 230:9–13.

Gemmell, M.A. 1976. Factors regulating tapeworm populations: estimations of the build-up and dispersion patterns of eggs after the introduction of dogs infected with Taenia hydatigena. Research in Veterinary Science.

Comments: Lines 288-291. It is incorrect to say that they would need carcass dissection plus PCR to differentiate animals infected with T. solium and T. hydatigena. For the vast majority of infections with either of these two parasites the identity of the species involved is clear based on the cysticercus morphology and site of encystment. It is recommended that the words ‘coupled with PCR’ be deleted.

Response: This has been noted and revised accordingly. Please note that Chembensofu et al. 2017 made the following observation: “An important finding from our study is the presence of T. solium cysticerci in the liver, spleen and lungs, locations that are not routinely included in full carcass dissections.” Hepatic T. solium cysticerci are not always easily differentiated from juvenile T. hydatigena cysticerci, hence the need for molecular confirmation.

Comments: Lines 296-298. Care needs to be exercised in commenting on the results of other publications which involved use of serology for determining the prevalence of T. solium. For many years the Ag-ELISA test was used and the fact that the test is not species specific was virtually ignored and the results assumed to relate to T. solium. Recent detailed evaluations of the test’s performance have revealed that there is a high proportion of animals that are positive in the test that have neither T. solium nor T. hydatigena infected. For example, Chembensofu et al 2017 (DOI 10.1186/s13071-017-2520-y) found that from 30 pig carcasses that were fully dissected and found negative for T. solium, the Ag-ELISA returned ten positives, of which two were infected with T. hydatigena. That is 8/30 (26.6%) animals that had neither T. solium nor T. hydatigena infection tested false positive in Ag-ELISA. With a false positive rate of this magnitude, it is not inconceivable that all 9 Ag-ELISA positive animals from the 126 that were tested in the study described in this manuscript may have no Taenia infection at all…

Response: Noted. The authors agree with the reviewer that cross-reactivity between T. hydatigena and T. solium remains a challenge. The authors discussed this in more details in the introduction and discussion sections of the manuscript. In the same study cited by the reviewer (Chembensofu et al 2017), the authors indicated that the sensitivity of the Ag-ELISA to detect carcasses with viable cysticerci (1 or more) was estimated at 91% (95% CI: 71–99%), and increased further to 100% (95% CI: 75–100%) for carcasses with at least 10 viable cysticerci. Therefore, the validity of the results of this study still remains as the objective was to investigate the prevalence of T. solium but prevalence of Taenia species. We agree that the study of Chembensofu et al. 2017 would indicate -the existence of ‘transient’ antigens. It is indeed possible that the Ag-ELISA detects infections at the early stages of infection (Deckers et al., 2008), before establishment of cysticerci, and that in some cases infections are ‘aborted’, i.e. that infection do not result in establishment of cysticerci because of immune-driven interruption, of the infection.

Deckers N., Kanobana K., Silva M., Gonzalez A.E., Garcia H.H.,Gilman R.H., Dorny P. (2008). Serological responses in porcine cysticercosis:A link with the parasitological outcome of infection. International Journal for Parasitology 38, 1191–1198

Comments: Lines 352-353. The sharing of the environment of children and pigs with dogs has nothing whatsoever to do with transmission of T. solium.

Response: This has been noted and revised accordingly.

Comments: Lines 400-401. It is difficult to understand which authors did what – it appears that the letters refer to authors’ Christian names. It is suggested to use all the authors’ initials.

Response: This has been noted and revised accordingly. 

Reviewer 2

Comments: Somewhere, in the introduction (lines 64 – 66) and in the discussion (lines 280-283) the authors clearly indicate that similar studies have been done in the same area. In my opinion, results of an undertaking reported here are just a duplication of something which has already been reported and doesn’t add any value. 

Response: The authors note the concerns raised by the reviewer and have revised the manuscript accordingly. Please see below. 

“There is evidence of Taenia spp. circulating in pigs in Gauteng, with 14% pigs reported to be infected in selected areas [15]. In 2016 reports of illegally slaughtered pigs for human consumption and high number of pigs with possible T. solium infection from the areas under study surfaced. Furthermore, pig farmers in these areas were reported to have had poor husbandry practices and the farms were in close proximity to human settlements. In view of this, this study estimated the seroprevalence of active Taenia spp. in domestic pigs in the selected areas of Gauteng and identified factors associated with infections and transmission based on knowledge and farming practices of pig owners. We hypothesize that Taenia spp. are circulating among pigs in township areas under study and that factors associated with the burden of Taenia spp. exist in these areas. “

TITLE

Comments: The authors need to recast the title, they could write it better than the way it is currently

Response: The title has been revised to read as “Knowledge, practices and seroprevalence of Taenia species in smallholder farms in Gauteng, South Africa”

ABSTRACT

Comments: Line 15-16: The sentence “Poor agricultural practices, and sanitary practices, and lack of…………” should be re-written to reduce the number of “ands”

Response: Changes made 

Comments: Line 22-25: The sentence “Descriptive statistics were run, and logistic regression models were used to assess the relationship between socio demographic and pig farming conditions and knowledge of porcine cysticercosis and neurocysticercosis.” should be re-written to reduce the number of “ands”

Response: The sentence has been rephrased to read as “Logistic regression models were used to assess the relationship between predictor variables and the outcome variables knowledge of porcine cysticercosis or knowledge of neurocysticercosis.”

INTRODUCTION

Comments: Lines 64 – 66: the authors could not clearly indicate the rationale of conducting the study

Response: The rationale of the study has been revised to read as “There is evidence of Taenia spp. circulating in pigs in Gauteng, with 14% pigs reported to be infected in selected areas [15]. In 2016, reports of illegally slaughtered pigs for human consumption and high number of pigs with possible T. solium infection from areas under study surfaced. Furthermore, pig farmers in these areas were reported to have had poor husbandry practices and the farms were in close proximity to human settlements. In view of this, this study estimated the seroprevalence of active Taenia spp. in domestic pigs in the selected areas of Gauteng and identified factors associated with infections and transmission based on knowledge and farming practices of pig owners. We hypothesize that Taenia spp. are circulating among pigs in township areas under study and that factors associated with the burden of Taenia spp. exist in these areas.”

MATERIALS AND METHODS

Comments: In line 86 the authors indicate the use of mixed method which are not evident in the subsequent part of the study design.

Response: We thank the reviewer for the comments. There is evidence of both qualitative and quantitative approach in this study. The selection of the study areas is based on the reports by veterinary extension services and the challenges they had identified in the areas understudy. Therefore, the qualitative approach formed the basis of this study. Furthermore, both positivism and non-positivism approach although not explicitly outlined there are in line with the approach used by the authors. The positivism approach refers to quantifying the level of Taenia species in pigs based on the serological tests. The non-positivism is in the knowledge and practises of pig farmers in relation to cysticercosis and taeniasis. 

Kaur, M., 2016. Application of mixed method approach in public health research. Indian Journal of Community Medicine: Official Publication of Indian Association of Preventive & Social Medicine, 41(2), p.93.

Bryman, A., 2006. Integrating quantitative and qualitative research: how is it done? Qualitative research, 6(1), pp.97-113.

Palinkas, L.A., Aarons, G.A., Horwitz, S., Chamberlain, P., Hurlburt, M. and Landsverk, J., 2011. Mixed method designs in implementation research. Administration and Policy in Mental Health and Mental Health Services Research, 38(1), pp.44-53.

Comments: In line 86-87 the authors state that they used purposive sampling to identify farms and farmers to participate in this study, they should clearly explain/reason justification for this.

Response: We thank the reviewer for the comments. It is possible that the reviewer might have missed this, the reasons for this approach are outlined in the study area and study populations section of the manuscript. “The selection of this type of sample is based on the elements possessing one or more attributes such as, known exposure to a risk factor or a specific disease status. This approach is often used in observational analytic studies. If a random sample is drawn from all sampling units meeting the study criteria, then it becomes a probability sample from the subset of the target population.” Dohoo, I.R., Martin, W. and Stryhn, H.E., 2003. Veterinary epidemiologic research.

Please see below

Study Area

Response: These areas were selected based on reports by veterinary extension services of poor pig farming husbandry practices and proximity between farms and human settlements. 

Study population 

Response: All pigs within the selected farms were tested provided that they met the following criteria i) Not pregnant, ii) older than 6 months, and iii) apparently healthy, therefore, making this step a probability sample [34]. The farmers were selected on the basis of known informal pig farming system and known risk factors for T. solium [27]. A total of 56 farmers were approached through communication with the State Veterinarians and Animal Health Technicians (AHT) from the Department of Agriculture and Rural Development in Gauteng Province. The inclusion criteria for the farmers were as follows: i) must agree to be part of the study, ii) must be 18 years or older, and iii) must be involved in pig farming as an owner or employee. 

Comments: Line 87-90: the authors explain the selection procedure of pigs to be involved in the study and they eventually refer to the step as being probability, THIS IS SERIOUS.

Response: “The selection of this type of sample is based on the elements possessing one or more attributes such as known exposure to a risk factor or a specific disease status. This approach is often used in observational analytic studies. If a random sample is drawn from all sampling units meeting the study criteria, then it becomes a probability sample from the subset of the target population.” Dohoo, I.R., Martin, W. and Stryhn, H.E., 2003. Veterinary epidemiologic research.

Comments: Line 87-88: the authors indicate that all pigs that met criteria in all the selected farms were sampled, what was the essence of calculating the sample size then?

The sample size was calculated in order to have an estimate of the number of animals to be sampled provided that the population size was large enough. At the time of the design of the study, information on pig populations in township areas under study was not available. 

Comments: The authors also need to state the number of farms from which pigs were sampled;

Response: The reviewer might have missed it, it is highlighted in line 88.

Comments: Line 90-95: the authors explain the selection procedure of farmers. It is not clear whether these were the owners of the farms where pigs were sampled or otherwise.

Response: The reviewer might have missed it, it is highlighted in line 88.

Comments: Line 98: the authors indicate that the calculated sample size was 185 pigs, but in Line 101: they indicate they sampled 126 pigs, this needs explanation.

Response: Although this was the estimated sample size, only 126 pigs met the criteria stated in line 90-91.

Comments: Lines 103-114: The authors do not consider “Blood collection and processing” and “Serological analysis” as part of data collection, can they give a reason to this?

Response: This has been corrected based in line with the comments of the reviewer. 

Comments: Lines 128-131: The pretesting of the questionnaire doesn’t seem to make sense as the population to which it was tested is very different in characteristics from farmers. The authors need to defend their decision.

Response: As stated in line 145-149, pre-testing included all individuals that were representative of the target population with input from experts in the field.

Comments: Line 150: The sentence “The open-ended questions were analyzed using thematic analysis” should be cancelled.

Response: Change made

RESULTS

Comments: Generally, this section is poorly organized, the flow is not good

Response: We thank the reviewer for the comments. However, it would have been valuable if the reviewer had provided guidance to which part of the results the authors need to address. 

In several occasions there is double presentation of results in text and in table

Response: The text provides a summary of the important findings from the results of the study. We would appreciate if the reviewer could provide guidance to which part of the results were repeated so that we are able to correct them. 

DISCUSSION

Comments: Generally, the discussion is poorly organized

Response: We thank the reviewer for the comments. However, it would have been valuable if the reviewer had provided guidance to which part of the discussion the authors need to attend. We look forward to this guidance from the reviewer.

Comments: References “7” cited in line 280 and reference “36” cited in line 283 indicate that similar works were conducted in the same area. This is a weakness as resources devoted in this study could be deployed elsewhere

Response: The authors note the concerns raised by the reviewer and have revised the manuscript accordingly. Please see below. There is evidence of Taenia spp. circulating in pigs in Gauteng, with 14% pigs reported to be infected in selected areas [15]. In 2016 reports of illegally slaughtered pigs for human consumption and high number of pigs with possible T. solium infection from areas under study surfaced. Furthermore, pig farmers in these areas were reported to have had poor husbandry practices and the farms were in close proximity to human settlements. In view of this, this study estimated the seroprevalence of active Taenia spp. in domestic pigs in the selected areas of Gauteng and identified factors associated with infections and transmission based on knowledge and farming practices of pig owners. We hypothesize that Taenia spp. are circulating among pigs in township areas under study and that factors associated with the burden of Taenia spp. exist in these areas.

CONCLUSIONS

Comments: The third word in line 385 (training) does sound good to me as it is linked to imparting farmers knowledge on epidemiology of the parasite

Response: We note the comments of the reviewer.

Ethical considerations

Comments: Line 389-390: Check this “The Animal Ethics Committee Research Ethics Committee”

Response: Correction made 

Comments: Line 393-394: I am doubting on the statement that; “Signed informed consent was obtained from each respondent before the questionnaires were administered”. This implies that all farmers could read and write

Response: All farmers could sign, and the study was explained in their language of comprehension before conducting the study.

Acknowledgements

Comments: Line 396: The statement “The author would like to acknowledge the Gauteng……….” Indicates that this manuscript is single authored

Response: Correction made 

 

Reviewer 3

Comments: All the manuscript always refers to Taenia spp, but since all the study was performed in pigs and since it also states human neurocysticercosis, I recommend to change spp for solium, mainly because it is more accurate even though the species was not identified but Taenia saginata is only found in cattle and it does not cause neurocysticercosis in humans and using T. solium more precise in educational terms. An exception is lines 290-291

Response: We thank the reviewer for the comments; however, we prefer to leave the title as is due to the fact that the test used in this study does not differentiate T. hydatigena from T. solium 

Comments: The authors should include the concept of KAP because they use it but it is not included as a specific issue such as in lines 117, 127, 157.

Response: Changes have been made in the introduction section of the manuscript in line with the comments of the reviewer. Please see below 

“Similarly, poor sanitary and hygiene conditions and free-range systems have been associated with increased risk of Taenia spp. in pig populations [4,21–23]. In addition, the education level [24,25], knowledge on livestock management [26,27], and poor farming practices have been associated with high prevalence of Taenia spp. infection in pig populations [29,57].”

Comments: The reference to the statistic test cited on lines 182-183 should be provided.

Response: Noted. Addressed

Comments: Line 210 does not state pigs in numbers between 10 and 20. The authors should add: 11-20 pigs, 48%

Response: Noted. Addressed

Comments: Line 228, if not plain water, what other source/type of water did the farmers use? 

Response: The farmers used plain water; correction has been made.

Comments: Table 3. farming purpose “all three” should be moved two lines below

Response: Noted. Addressed

Line 310 eliminate “of age”

Response: Noted. Addressed

Comments: The subject of KAP should also be included in the paragraph that starts on line 359

Response: It is not clear what the reviewer is requesting. Once we have received guide, we will make the changes as requested. 

Comments: In acknowledgments I would add to the farmers that participated in the study

Response: Noted. Addressed

---

## [Decision Letter · Decision Letter 1]

15 Sep 2020

PONE-D-20-14308R1

Knowledge, practices and seroprevalence of Taenia species in smallholder farms in Gauteng, South Africa

PLOS ONE

Dear Dr. Qekwana,

Thank you for submitting your manuscript to PLOS ONE. After careful consideration, we feel that it has merit but does not fully meet PLOS ONE’s publication criteria as it currently stands. Therefore, we invite you to submit a revised version of the manuscript that addresses the points raised during the review process.

Many thanks for submitting your manuscript to PLOS One

It was reviewed by the same experts in the field, and they have requested some more minor changes be made prior to acceptance.

If you could make these changes and write a response to reviewers, that will greatly expedite revision upon resubmission

I wish you the best of luck with your changes

Hope you are keeping safe and well in these difficult times

Thanks

Simon

We look forward to receiving your revised manuscript.

Kind regards,

Simon Clegg, PhD

Academic Editor

PLOS ONE

Reviewers' comments:

Reviewer's Responses to Questions

**Comments to the Author**

1. If the authors have adequately addressed your comments raised in a previous round of review and you feel that this manuscript is now acceptable for publication, you may indicate that here to bypass the “Comments to the Author” section, enter your conflict of interest statement in the “Confidential to Editor” section, and submit your "Accept" recommendation.

Reviewer #1: (No Response)

Reviewer #3: All comments have been addressed

2. Is the manuscript technically sound, and do the data support the conclusions?

Reviewer #1: Yes

Reviewer #3: Yes

3. Has the statistical analysis been performed appropriately and rigorously? 

Reviewer #1: I Don't Know

Reviewer #3: Yes

4. Have the authors made all data underlying the findings in their manuscript fully available?

Reviewer #1: Yes

Reviewer #3: Yes

5. Is the manuscript presented in an intelligible fashion and written in standard English?

Reviewer #1: Yes

Reviewer #3: Yes

6. Review Comments to the Author

Reviewer #1: I am disappointed with the authors' responses to several of the comments I made on the manuscript. In many places they choose to argue, in effect, that I was wrong. This would be entirely acceptable if they were correct and I was not, but when the opposite is the case, it is galling.

For example, in response to my indicating that the value of the manuscript would have been enhanced had they purchased the serologically positive pigs and shown that they were actually infected with T. solium, the authors simply provide a list of excuses as to why they did not do it. The fact they did not do it is the only thing that is relevant, not why they could not, or choose not.

In response to my criticism about the lack of specificity of the serological test uses, the authors respond by citing a reference that specifies the test had a specificity of 67% for T. solium. That is exactly my point. That is a very poor specificity, and there is certainly a possibility that all 9 of the 126 animals that the authors found positive may well have had no T. solium infection at all. As the authors choose to cite two papers about the performance of the B158/B60 Ag-ELISA in relation to diagnosis of T. solium infection in pigs, and recognise that the test is also positive in pigs infected with T. hydatigena, for balance they should add the test’s sensitivity for diagnosis of T. hydatigena infection. This information is in the literature and I believe they will find that the test’s performance for T. hydatigena is just a good as it is for T. solium.

The authors’ response to my statement that the commercial test they used is not the same as the in-house test they cite in relation to the test’s performance, they firstly accept this is true but make excuses about what they perceive as small differences in the methodology used in the kit form of the test. They are not small differences, and they certainly have potential to affect the test’s performance.

The authors’ response to my comment that it is not necessary to invoke the use of PCR to differentiate T. solium and T. hydatigena cysts was to cite Chembensofu et al 2017 who claimed to have found numerous instances of T. solium cysts in the liver and lungs of pigs in Zambia and that Chembensofu et al validated their results by PCR. It is interesting to note that Chembensofu et al provided no actual data to support their extraordinary claims about cysts in ‘unusual’ locations, nor any information about what controls were used in their PCRs. The authors of the present manuscript seem unaware of the publication by Gauci et al 2019 https://doi.org/10.1371/journal.pntd.0007408 who, in response to the Chembensofu et al publication, investigated ‘lesions’ in the tissues of pigs from T. solium endemic areas of Nepal and Uganda, such as the liver and lung, that could possibly have been confused with being caused by T. solium. No T. solium cyst was found in any organ other than in the striated muscles and the brain. Numerous ‘lesions’ in other body organs were confirmed as being caused by a T. hydatigena or nematodes, as well as other causes. Gauci et al presented actual PCR results, and specified that tissue from the same animal and organ where a suspect lesion was found, but containing no lesion, was used as a negative control in the PCRs. I make two points here. Firstly, by statement that PCR is not required to differentiate T. solium and T. hydatigena cysts stands as correct. Secondly, it is annoying that the authors choose to argue against the point I made in my original review in relation to this matter, when they are clearly ignorant of the recent relevant literature. This is especially surprising because the two Belgian-based authors have a long history in the T. solium field.

The changes that have been made to the manuscript are adequate in relation to the (lack of) specificity of the serological studies they performed and the limitations this imposes on the interpretation of their results. I reiterate what I said in my original review, which was that the study would have been so much more meaningful if they had obtained actual evidence that any of the pigs were infected with T. solium. The authors’ reliance on poor quality serology was a major flaw in the study design and has resulted in a substantial reduction in the potential value of the study.

Reviewer #3: (No Response)

7. PLOS authors have the option to publish the peer review history of their article (what does this mean?). If published, this will include your full peer review and any attached files.

Reviewer #1: No

Reviewer #3: No

---

## [Author Response · Author response to Decision Letter 1]

27 Nov 2020

Comment 1: I am disappointed with the authors' responses to several of the comments I made on the manuscript. In many places they choose to argue, in effect, that I was wrong. This would be entirely acceptable if they were correct and I was not, but when the opposite is the case, it is galling.

For example, in response to my indicating that the value of the manuscript would have been enhanced had they purchased the serologically viable pigs and shown that they were actually infected with T. solium, the authors simply provide a list of excuses as to why they did not do it. The fact that they did not do it is the only thing that is relevant, not why they could not or choose not. 

Response: The authors sincerely apologize for the miscommunication. It was not our intention to be argumentative, but our intentions were to highlight the challenges we phased during the implementation of the study. We concur that the quality of this study would have been enhanced if serologically viable pigs were purchased and the body parts excised to confirm the infection using PCR-RFLP. In trying to address the concern of the reviewer we have highlighted these limitations in lines 376 to 381.

Comment 2: In response to my criticism about the lack of specificity of the serological test uses, the authors respond by citing a reference that specifies the test had a specificity of 67% for T. solium. That is exactly my point. That is a very poor specificity, and there is certainly a possibility that all 9 of the 126 animals that the authors found positive may well have had no T. solium infection at all. As the authors choose to cite two papers about the performance of the B158/B60 Ag-ELISA in relation to diagnosis of T. solium infection in pigs, and recognise that the test is also positive in pigs infected with T. hydatigena, for balance they should add the test’s sensitivity for diagnosis of T. hydatigena infection. This information is in the literature and I believe they will find that the test’s performance for T. hydatigena is just a good as it is for T. solium. 

Response: The serological assay used (B158/B60 Ag-ELISA) detects circulating antigens of Taenia spp. which indicates infection with either Taenia hydatigena or Taenia solium. For this reason, one of the aims of this paper was to study the seroprevalence of Taenia spp., without specifying. In the Chembensofu et al 2017 paper, the specificity (se) and sensitivity (sp) of the Ag-ELISA for the diagnosis of T. solium were estimated at 67% and 68%, respectively, while the sp and se of this test for diagnosing T. hydatigena infection were 49% and 86%, respectively when considering the results of the 68 dissected pigs. We agree with the reviewer that it cannot be excluded that all of the 9 of the 126 animals that were found positive by Ag-ELISA were infected with T. hydatigena. 

Comment 3: The authors’ response to my statement that the commercial test they used is not the same as the in-house test they cite in relation to the test’s performance, they firstly accept this is true but make excuses about what they perceive as small differences in the methodology used in the kit form of the test. They are not small differences, and they certainly have potential to affect the test’s performance.

Response: Thank you for this remark. There are indeed differences in test conditions between the in house and the commercial Ag-ELISA. The commercial test has reduced the number of steps in the ELISA, modified the substrate and uses a different method for cut off calculation, in order to make the test more user friendly. Nevertheless, one of the authors of this paper was involved in the validation of the commercial test, during which the results of a substantial number of well-documented serum samples were compared with those of the in house test and test performance was not significantly different between the two test formats.

Comment 4: The authors’ response to my comment that it is not necessary to invoke the use of PCR to differentiate T. solium and T. hydatigena cysts was to cite Chembensofu et al 2017 who claimed to have found numerous instances of T. solium cysts in the liver and lungs of pigs in Zambia and that Chembensofu et al validated their results by PCR. It is interesting to note that Chembensofu et al provided no actual data to support their extraordinary claims about cysts in ‘unusual’ locations, nor any information about what controls were used in their PCRs. The authors of the present manuscript seem unaware of the publication by Gauci et al 2019 https://doi.org/10.1371/journal.pntd.0007408 who, in response to the Chembensofu et al publication, investigated ‘lesions’ in the tissues of pigs from T. solium endemic areas of Nepal and Uganda, such as the liver and lung, that could possibly have been confused with being caused by T. solium. No T. solium cyst was found in any organ other than in the striated muscles and the brain. Numerous ‘lesions’ in other body organs were confirmed as being caused by a T. hydatigena or nematodes, as well as other causes. Gauci et al presented actual PCR results and specified that tissue from the same animal and organ where a suspect lesion was found, but containing no lesion, was used as a negative control in the PCRs. I make two points here. Firstly, by statement that PCR is not required to differentiate T. solium and T. hydatigena cysts stands as correct. Secondly, it is annoying that the authors choose to argue against the point I made in my original review in relation to this matter, when they are clearly ignorant of the recent relevant literature. This is especially surprising because the two Belgian-based authors have a long history in the T. solium field.

Response: The authors are aware of the paper of Gauci et al. 2019, and it is indeed interesting to observe these contrasting results. We agree that some studies, including the Gauci et al 2019 study have performed dissection and have not found cysts in other organs except for those routinely examined for T. solium infections such as striated muscles and brain. However, the presence of T. solium cysticerci in the liver has also been found in pigs in Burkina Faso (Dermauw et al 2016 doi: 10.1016/j.vetpar.2016.10.022). Two of the authors of the present paper were involved in the pig dissections of either the Dermauw et al. or in both the Dermauw et al. and the Chembensofu et al papers. 

In both the Dermauw and Chembensofu studies, suspected lesions were excised from the liver and other organs and transferred to individual tubes containing ethanol to avoid cross contamination of genetic material. DNA extractions and PCR’s were performed in the molecular lab of ITM Antwerp, according to SOP’s and a stringent laboratory quality system, including the use of negative and positive controls. 

Following the findings of T. solium cysticerci in the liver and other organs, we consider the confirmation of suspected lesions by molecular methods important in the validation or identification of Taenia spp. cysts to nullify dubious cysts and confirm suspected lesions.

Comment 5: The changes that have been made to the manuscript are adequate in relation to the (lack of) specificity of the serological studies they performed and the limitations this imposes on the interpretation of their results. I reiterate what I said in my original review, which was that the study would have been so much more meaningful if they had obtained actual evidence that any of the pigs were infected with T. solium. The authors’ reliance on poor quality serology was a major flaw in the study design and has resulted in a substantial reduction in the potential value of the study.

Response: Thank you for your appreciation of the implemented changes. With regard to the comment on the use of serology, kindly refer to our response on Comment 1.

---

## [Editor Report · Decision Letter 2]

3 Dec 2020

Knowledge, practices and seroprevalence of Taenia species in smallholder farms in Gauteng, South Africa

PONE-D-20-14308R2

Dear Dr. Qekwana,

We’re pleased to inform you that your manuscript has been judged scientifically suitable for publication and will be formally accepted for publication once it meets all outstanding technical requirements.

Kind regards,

Simon Clegg, PhD

Academic Editor

PLOS ONE

Additional Editor Comments:

Many thanks for resubmitting your manuscript to PLOS One

As you have addressed all the comments, and the manuscript reads well, I have recommended the manuscript for publication

You should hear from the Editorial office soon

It was a pleasure working with you, and I wish you the best of luck for your future research

Hope you are keeping safe and well in these difficult times

Thanks

Simon

---

## [Editor Report · Acceptance letter]

9 Dec 2020

PONE-D-20-14308R2 

Knowledge, practices and seroprevalence of *Taenia*species in smallholder farms in Gauteng, South Africa 

Dear Dr. Qekwana:

I'm pleased to inform you that your manuscript has been deemed suitable for publication in PLOS ONE. Congratulations! Your manuscript is now with our production department. 

Kind regards, 

on behalf of

Dr. Simon Clegg 

Academic Editor

PLOS ONE